# A metastable liquid melted from a crystalline solid under decompression

Chuanlong Lin[1], Jesse S. Smith[1], Stanislav V. Sinogeikin[1], Yoshio Kono[1], Changyong Park[1], Curtis Kenney-Benson[1] & Guoyin Shen[1]

A metastable liquid may exist under supercooling, sustaining the liquid below the melting point such as supercooled water and silicon. It may also exist as a transient state in solid–solid transitions, as demonstrated in recent studies of colloidal particles and glass-forming metallic systems. One important question is whether a crystalline solid may directly melt into a sustainable metastable liquid. By thermal heating, a crystalline solid will always melt into a liquid above the melting point. Here we report that a high-pressure crystalline phase of bismuth can melt into a metastable liquid below the melting line through a decompression process. The decompression-induced metastable liquid can be maintained for hours in static conditions, and transform to crystalline phases when external perturbations, such as heating and cooling, are applied. It occurs in the pressure–temperature region similar to where the supercooled liquid Bi is observed. Akin to supercooled liquid, the pressure-induced metastable liquid may be more ubiquitous than we thought.

[1] HPCAT, Geophysical Laboratory, Carnegie Institution of Washington, Argonne, Illinois 60439, USA. Correspondence and requests for materials should be addressed to G.S. (email: gshen@ciw.edu).

A supercooled liquid may be obtained by cooling a stable liquid below the melting line where the crystalline phase is stable[1,2]. The supercooled region (that is, temperature and pressure conditions where the supercooled liquid exists) is highly related to the kinetic energies of nucleation and grain growth, and is therefore sensitive to external perturbations, for example, impurity, vibration, heating and/or cooling[3]. In contrast, a crystalline solid always melts into a liquid above the melting line[3], although the melting process may be affected by factors such as heating rate, impurities, particle size and shear stress. Recently, there has been a growing interest[4–11] in studying whether a crystalline solid may directly melt into a metastable liquid below melting line ($T_m$) and the mechanisms underlying it.

Levitas et al.[4,5] proposed a transient melting process that occurs in a solid–solid phase transformation at temperatures significantly below the melting temperature due to internal stress. Recently such transient liquid has been experimentally observed using polymeric colloidal systems[6–8], where the particles mimic atoms. A two-step diffusive nucleation pathway was observed in a structural transition, that is, appearance of transient liquid first followed by its crystallization[6]. The occurrence of the transient liquid is attributed to a much smaller interfacial energy barrier at the solid/liquid interface than that at the solid/solid interface. It was further suggested that an intermediate liquid may exist in solid–solid transformations of metals and alloys, because the interfacial energy for solid/liquid interfaces ranges from 30 to $\sim 250$ mJ m$^{-2}$, much lower than that (500 to $\sim 1,000$ mJ m$^{-2}$) for the incoherent solid/solid interfaces[9,10]. Very recently, transient liquid was adopted in interpreting the fast differential scanning calorimetry data in a bulk metallic glass sample via rapid heating[11]. In a study of pressure-induced melting of ice, endothermic responses indicated the coexistence of two different phases of supercooled liquid water[1,12,13]. All these imply the existence of a metastable liquid below $T_m$. However, the atomistic nature of a metastable liquid remains untested by in situ probes such as X-ray diffraction.

We here perform experiments on elemental bismuth (Bi) under hydrostatic conditions in diamond anvil cells (DACs) using in situ X-ray diffraction. We observe that a crystalline solid phase of Bi can directly melt into a metastable liquid below the melting line. The metastable liquid can be kept for several hours at static condition until external perturbations are applied such as heating or cooling, resulting in transformation to crystalline phases.

## Results

**Phase diagram.** Bismuth has a complex phase diagram, exhibiting several polymorphs and a V-shape melting curve (Supplementary Fig. 1)[14]. At ambient conditions, the rhombohedral structure (Bi-I) is the stable phase with $Z = 2$ and space group $R\bar{3}m$ (Supplementary Fig. 1). Bi-I melts at $\sim 544$ K at ambient pressure[14]. The structure of Bi-I can be viewed as a slightly distorted primitive cubic structure[15]. Similar to ice Ih, Bi-I has a negative Clausius–Clapeyron melting slope. Under compression at room temperature, Bi-I transforms to Bi-II with volume collapse of $\sim 4.7\%$ at $\sim 2.5$ GPa (ref. 14). Bi-II has a monoclinic structure (Supplementary Fig. 1)[16]. The layer structure of Bi-II is similar to Bi-I, and can be described as a heavily distorted primitive cubic array[15]. Upon further compression, Bi-II transforms to Bi-III at $\sim 2.8$ GPa (ref. 17), a tetrahedral host–guest structure (Supplementary Fig. 1). Bi-II′ was found at 1.9 GPa and 463 K and exists in a small pressure–temperature region[18]. It has the $\beta$-tin type structure with $a = 6.210$ Å and $b = 3.311$ Å. The triple points are at 1.65 GPa and 465 K for I–II′-liquid, 1.96 GPa and 465 K for II–II′-liquid, and 2.1 GPa and 464 K for II–IV-liquid[14,18].

**a**
Compression at 489 K
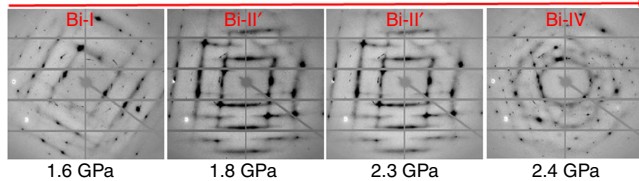

**b**
Decompression at 489 K
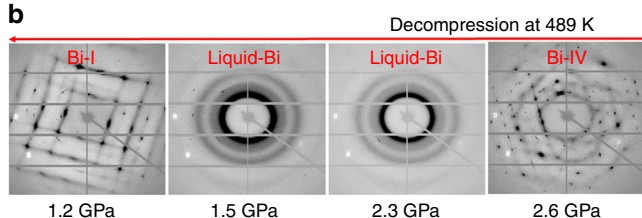

**Figure 1 | Diffraction images at 489 K under compression and decompression pathways.** Neon serves as pressure medium to provide hydrostatic condition. The detailed phase transition sequences and indexes of crystalline phases are shown in Supplementary Fig. 2. (**a**) Bi sample is compressed from 1.0 to 3.2 GPa (right red arrows). Bi-I phase transforms to Bi-II′ at $\sim 1.8$ GPa, followed by II′-to-IV transition at $\sim 2.4$ GPa. (**b**) Bi sample is decompressed from 3.2 to 1.0 GPa (left red arrows). Bi-IV melts into liquid Bi at $\sim 2.3$ GPa, followed by crystallization of liquid Bi at $\sim 1.2$ GPa.

Bi-II′ and Bi-II have a flat melting line. Above 465 K, liquid Bi transforms to Bi-IV under compression[14]. The high-pressure and high-temperature phase of Bi-IV has an orthorhombic $oC16$ structure with $a = 11.191(5)$ Å, $b = 6.622(1)$ Å and $c = 6.608(1)$ Å (ref. 19), as shown in Supplementary Fig. 1.

**Decompression-induced liquid under hydrostatic condition.** Under an isothermal compression at 489 K (ref. 14), Bi should transform from Bi-I to a liquid and then to Bi-IV, or vice versa under decompression. This is precisely what we observe in Bi samples (Alfa Aesar, purity of 99.99%) at 489 K during the compression and decompression pathways in DACs without pressure medium or using a soft solid medium of NaCl (Supplementary Fig. 2). However, when we perform experiments on Bi in a hydrostatic neon medium in a pressure range of 1.0–3.5 GPa at 489 K (see the sample loading in Supplementary Fig. 3), we find that the Bi sample undergoes a crystalline–crystalline transition from Bi-I to Bi-II′ at $\sim 1.8$ GPa under compression at 489 K, followed by another solid–solid transformation from Bi-II′ to Bi-IV at $\sim 2.4$ GPa (Fig. 1a). Apparently, the shear stress may have played an important role in lowering melting temperature of Bi-I, as reported in other materials[20–24]. The use of liquid neon in this study provides excellent hydrostatic condition, thus revealing the true melting temperature higher than 489 K. Upon decompression at 489 K, Bi-IV melts into a liquid at $\sim 2.3$ GPa, and crystallizes into Bi-I at $\sim 1.2$ GPa under further decompression (Fig. 1b). From optical imaging measurements, it is clearly observed that there are morphology changes with sharp edges of irregular shapes in the phase transition under compression (Supplementary Movie 1), indicating a solid–solid transition, whereas the Bi sample displays liquid morphology during decompression (Supplementary Movie 2), indicating the melting upon decompression. We repeat this compression-decompression pathway five times, each with new fresh sample, at hydrostatic conditions using neon as a pressure medium. The Bi samples are repeatedly found to

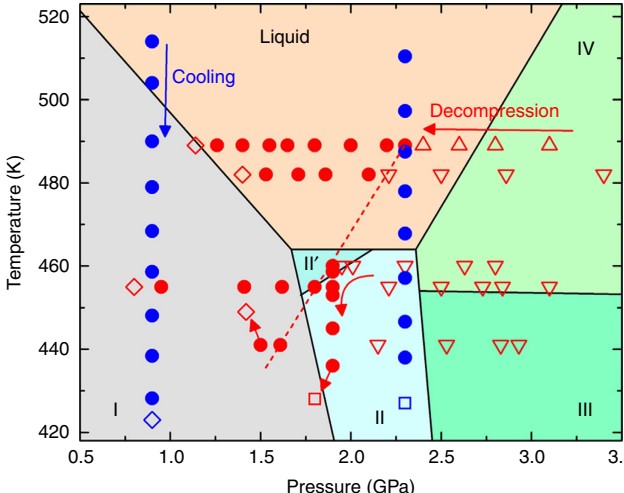

**Figure 2 | Experimental conditions where the metastable liquid is observed.** Bi-III′ below 489 K is obtained by compressing the DIL at 489 K, and then cooled down to lower temperatures. Under decompression (red left arrows) at 489, 482, 460, 455 and 441 K, Bi-III′ (red open down triangles) and Bi-IV (red open up triangles) melt into liquid Bi (red solid circles), followed by crystallization into Bi-I (red open diamonds). Upon compression, transition sequences of Bi-I→II (or II′)→IV (or mixture of IV + III) are observed (Supplementary Figs 2, 4, 7). Red dash line is a guide for eyes representing the decompression-induced melting line of Bi-III′ (or IV). At 460 K, Bi-III melts into the DIL at 1.9 GPa under decompression, and then the DIL is cooled (red down arrow) from 460 to 423 K at ∼1.8 GPa. The DIL crystallizes into Bi-II (red open squares) under cooling at 428 K with a slight change in pressure. At 441 K, Bi-III′ melts into liquid Bi at∼1.6 GPa. Then the DIL crystallizes into Bi-I upon heating (red up arrow) at 1.5 GPa from 441 to 449 K. Blue solid circles indicate liquid Bi under cooling (blue down arrow) from equilibrium liquid to supercooled liquid at ∼0.9 and 2.2 GPa, respectively. The supercooled liquid Bi crystallizes into Bi-I (blue open diamonds) at 0.9 GPa 423 K and into Bi-II (blue open squares) at 2.2 GPa 426 K, respectively.

undergo Bi-I→Bi-II′→ Bi-IV transitions upon compression from 1.0 to 3.2 GPa at 489 K, but display melting and crystallization (Bi-IV→liquid→Bi-I) upon decompression. The detailed phase transition sequences and the index of the crystalline phases are shown in Supplementary Fig. 4 and Supplementary Table 1.

The decompression-induced liquid (DIL) from Bi-IV at 489 K remains stable for several hours if pressure is maintained above 1.2 GPa. When pressure is increased, the DIL crystallizes into a new high-pressure phase at ∼3.1 GPa at 489 K, which can be indexed into a mixture of two body-centred-tetragonal phases (Supplementary Fig. 5 and Supplementary Table 1) with the lattice parameters close to those of the host and guest structures of Bi-III at ∼3 GPa and room temperature[17]. We denote this new phase as Bi-III′, a Bi-III-like host–guest structure. Indeed, there are similarities between diffraction patterns of Bi-III and Bi-III′ (Supplementary Fig. 5), for example, similar straight diffuse scattering lines, which come from the disordered guest chains[25–29]. As in the case for Bi-IV, we also find that Bi-III′ melts at ∼2.4 GPa upon decompression at 489 K.

The DIL not only appears at 489 K, but also is observed at lower temperatures. We performed compression–decompression experiments on Bi-III′ at different temperatures of 482, 460, 455 and 441 K, and observed decompression-induced melting of Bi- III′. Figure 2 summarizes the DIL melted from Bi-IV and Bi- III′ under hydrostatic conditions and different temperatures. Below 489 K, Bi-III′ is maintained while cooling it from 489 K to

corresponding temperatures. As found at 489 K, Bi-III′ transforms to the DIL upon decompression at 482, 460, 455 and 441 K (Supplementary Fig. 6 and Supplementary Table 2), followed by crystallization into Bi-I upon further decompression. In contrast, under compression at all these temperatures, we always observe solid–solid transitions following Bi-I→Bi-II (or II′)→Bi-IV (or mixture of IV + III).

It should be noted that the melting pressure under decompression shifts to lower pressures with decreasing temperature (Fig. 2). The decompression-induced melting line (red dash line in Fig. 2) is almost parallel to the previously reported melting line of Bi-IV obtained by piston-cylinder measurements[14]. At an even lower temperature of 423 K, Bi-III′ transforms to Bi-II at 2.2 GPa instead of melting, indicating the limited temperature range for the DIL. Overall, the DIL occurs in a similar region as that of supercooled Bi liquid. We performed two separate experiments by cooling the equilibrium liquid down below the melting line at 0.9 and 2.3 GPa, and find that the supercooled liquid Bi crystallizes at 423 and 426 K, respectively. Comparing the structures of the DIL and the supercooled liquid (Supplementary Figs 7 and 8), we find that the DIL has a similar structure to that of the supercooled liquid, reflected by the similarities of the structure factor $S(Q)$ and the pressure–temperature dependences of $Q_1$ and $Q_2/Q_1$ between them.

**Evidence of a metastable liquid.** Within a certain pressure–temperature region, we have observed two distinctly different phases: the crystalline solid (Bi-II or Bi-II′) on compression and the DIL on decompression. At least, one of them must be metastable. Interestingly, crystalline Bi-II (or II′) under compression at 489 and 482 K occurs above the reported $T_m$ of 464 K. Because shear stress could significantly lower melting temperature[20–24], the true melting temperature of Bi-II and Bi-II′ under hydrostatic condition may be higher than the reported 464 K. On the other hand, the lowest temperature where we observed the DIL is 441 K, far below the reported $T_m$. It is then likely that the DIL could be metastable. To further test the metastability of the DIL, we apply perturbations to the DIL by heating and cooling. When the DIL is heated from 441 to 449 K at 1.5 GPa, we find that the DIL crystallizes into Bi-I upon heating with a change of pressure from 1.5 to 1.4 GPa (Fig. 3a and integrated diffraction patterns in Supplementary Fig. 9). In a separate run, when the DIL is cooled from 460 to 428 K at ∼1.9 GPa (Fig. 3a and Supplementary Fig. 9), we find that the DIL crystallizes at 428 K, a temperature similar to the crystallization temperature of the supercooled liquid Bi (Fig. 2). Therefore, these perturbation tests support the metastable nature of the DIL.

**Discussion**
To understand how the decompression-induced metastable liquid can be formed below the thermal equilibrium melting line, we treat the transition process under decompression into two steps. The first step is the formation of a transient liquid at the onset of the phase transition from Bi-III′ or Bi-IV to low-pressure phases (Bi-II′ or Bi-II), followed by a sustaining process of the transient liquid similar to supercooled liquid. According to the classical nucleation theory, the probability of the homogeneous nucleation ($I$) is determined by the Arrhenius equation[10]:

$$I \sim \exp[-\Delta G^*/k_B T] \qquad (1)$$

where $\Delta G^*$ is the free energy barrier in the formation of nucleation, and can be expressed as:

$$\Delta G^* \sim \gamma^3/(\Delta G_V)^2 \qquad (2)$$

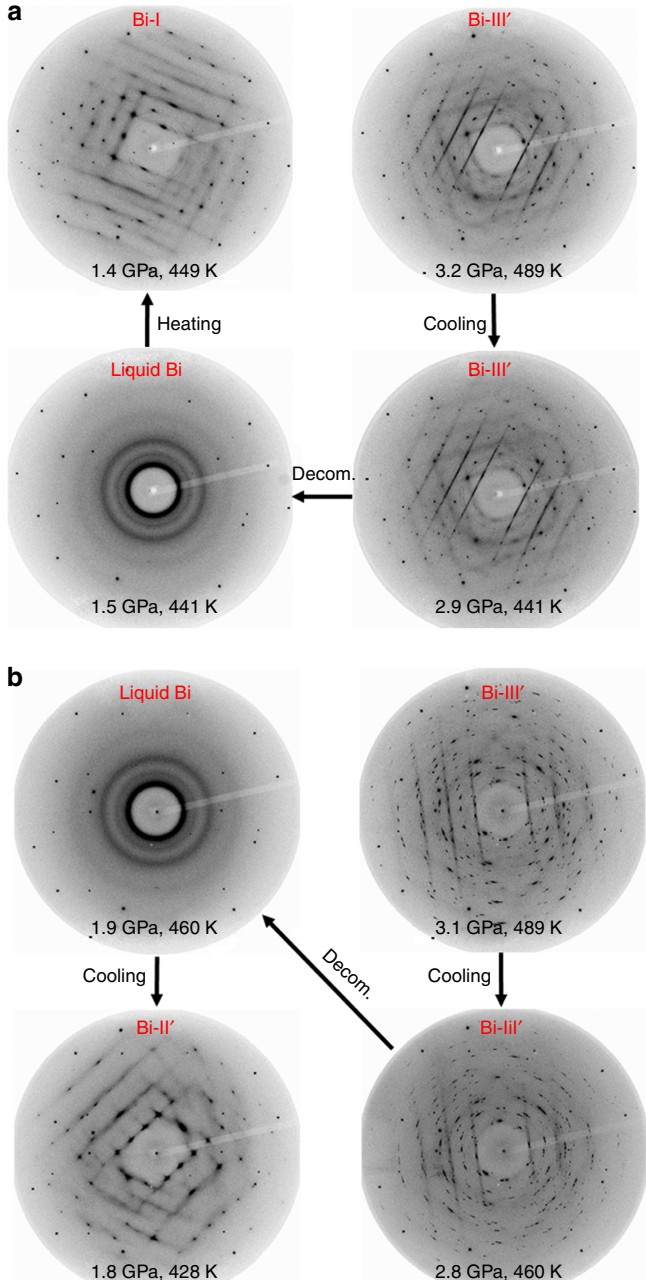

**Figure 3 | Crystallization of the decompression-induced liquid under heating and cooling.** (**a**) Bi-III′ is obtained by compressing the DIL at 489 K. The Bi-III′ then melts into a liquid state under decompression (decom.). The DIL later crystallizes into Bi-I at 449 K upon heating at 1.5 GPa. After crystallization, the pressure changes from 1.5 to 1.4 GPa. (**b**) The DIL at 1.9 GPa crystallizes into Bi-II under cooling at 428 K. The pressure changes from 1.9 to 1.8 GPa after crystallization.

where $\gamma$ and $\Delta G_V$ are the interfacial energy and the difference of Gibbs free energy between parent and new phases, which are closely correlated to structures[9,10]. $\Delta G_V$ is the thermodynamic driving force in the phase transitions.

In the first step, there are significant differences between the crystal structures of Bi-III′ (or Bi-IV) and Bi-II (or Bi-II′; Supplementary Fig. 1)[16–19], implying a large incoherent interfacial energy ($\gamma$(II/III′)) in II/III′ (or II/IV) interface[9]. From our measured $S(Q)$ and pair distribution function $G(r)$ of

the DIL (Supplementary Fig. 7), the nearest-neighbour coordination number (CN) of the DIL lies between those of Bi-II and Bi-III′ (or Bi-IV; Fig. 4a). It should be noted that the CN of the DIL increases with increasing pressure, with the CN close to that of Bi-II at low pressures and approaching that of Bi-III′ (or Bi-IV) at higher pressures. The structural feature of the DIL implies much lower interfacial energy $\gamma$(L/III′) in the liquid/III′ interface than that in the solid–solid Bi-II/III′ interface, that is, $\gamma$(L/III′) $\ll$ $\gamma$(II/III′)[9,30,31]. This is consistent with the previous studies, which show that $\gamma$ is $\sim$50–82 mJ m$^{-2}$ for the solid/liquid interface in Bi[32,33], at least twice smaller than that of the solid/solid interface[9,34]. According to equation (2), this will result in a smaller free energy barrier ($\Delta G^\star$(III′/L)) in the Bi-III′ to DIL transition than $\Delta G^\star$(III′/II) in the III′ to II transition, that is, $\Delta G^\star$(III′/L) < $\Delta G^\star$(III′/II). Thus, the probability of formation (equation (1)) of a transient liquid nuclei in III′ (or IV) will be larger than that of Bi-II (or Bi-II′; Fig. 4b). Therefore, the small interfacial energy is likely the main cause of the formation of the transient liquid for Bi in the first step[6,8].

Then why does the transient liquid persist over a long time without crystallization into Bi-II (or II′)? We attribute the preservation of the transient liquid to a process similar to supercooling, based on the following three arguments. (1) The DIL is observed in a pressure–temperature region similar to that of the supercooled liquid Bi (Fig. 2). (2) The structure of the DIL is the same as that of the supercooled liquid Bi. (3) Because the structure of B-II (or II′) is close to that of Bi-liquid; together with the negligible volume change between Bi-II and liquid Bi across the melting line, which is parallel to pressure (Fig. 2), the difference of free energies $\Delta G_V$(II/L) must be very small. The small denominator term $\Delta G_V$ in equation (2) leads to a large energy barrier in crystallization and a small probability for nucleation of Bi-II (Fig. 4b). The hydrostatic environment in our experiment may have played an important role in the preservation of the transient liquid. When the DIL is further decompressed into the pressure–temperature region where Bi-I is stable, the free energy difference between the DIL and Bi-I increases, resulting in a decrease of the energy barrier and an increase of probability in the nucleation of Bi-I (Fig. 4b). Eventually, the DIL crystalizes into Bi-I on decompression.

The observation of the DIL in Bi is conceptually consistent with the two-step nucleation mechanism in solid–solid phase transitions[6,7], that is, the formation of a transient liquid followed by a crystallization process. Here, rather than the nucleation and crystal growth, we observe the preserved transient liquid owing to a large kinetic energy barrier in crystallization. Our results indicate that a transient metastable liquid can be melted directly from a crystalline phase and preserved if the crystallization of the metastable liquid in the second step is suppressed. The decompression-induced metastable liquid may be ubiquitous in the first step and possibly observed in other metals, such as Na, K, Rb, Cs, Ba, Si and Ge, which, like Bi, have negative and positive melting lines and large structural differences between low-pressure phases and high-pressure phases. The preservation of the metastable liquid in Bi may be unique owing to the structural similarities in Bi-II and supercooled liquid Bi, which leads to the suppression of crystallization of the metastable liquid.

In conclusion, we observe a metastable liquid directly melted from a high-pressure crystalline phase in bismuth under decompression at hydrostatic condition by combining *in situ* synchrotron X-ray diffraction, high-temperature and high-pressure techniques. The decompression-induced metastable liquid occurs in the pressure–temperature region similar to where the supercooled liquid Bi is observed. Akin to supercooled liquid, the decompression-induced metastable liquid can persist

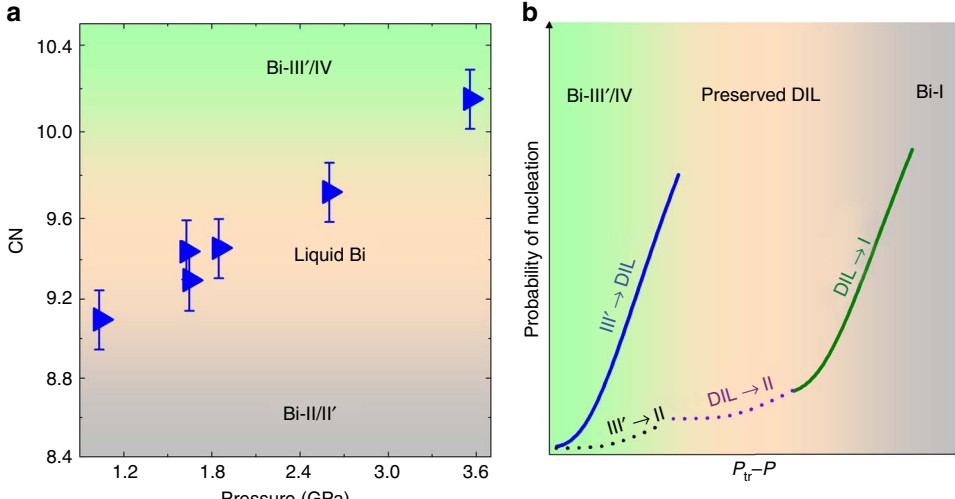

**Figure 4 | Coordination number of Bi in metastable liquid and probability of nucleation.** (**a**) Change of the nearest-neighbour coordination number (CN) of liquid Bi with pressure. The CN (blue solid right triangle) is calculated using the pair distribution function of liquid, assuming that the density of liquid is equal to that of Bi-II at corresponding pressure–temperature conditions. The uncertainties of CN are estimated from the errors in fitting the first sharp peak in the pair distribution functions of the liquid. The CN is close to that of Bi-II at low pressures, while the CN increases with pressure and approaches to that of Bi-IV at high pressures. (**b**) Nucleation probability in the phase transitions as a function of $P_{tr} - P$ under decompression, where $P_{tr}$ is thermodynamic equilibrium pressure of Bi-III' (or IV) and II. The probability is proportional to $\exp[-\Delta G^*/k_B T]$, where $\Delta G^*$ is the free energy barrier in the formation of nucleation, and is proportional to $\gamma^3/(\Delta G_V)^2$; $\gamma$ and $\Delta G_V$ are the interfacial energy and difference of Gibbs free energy between parent and new phases, which are closely correlated to structures. In the first step, Bi-III' to DIL transition has large probability in formation of liquid (blue solid line) compared with that of Bi-II in Bi-III to Bi-II transition (black dash line), owing to much smaller interfacial energy in solid/liquid interfaces. In the second step, the large kinetic energy barrier in crystallization of the transient liquid suppresses the formation of Bi-II (purple dash line), eventually resulting in the preservation of the DIL. Upon further decompression, probability of nucleation of Bi-I (olive solid line) increases due to the increase in difference of Gibbs free energies between DIL and Bi-I.

over a long time until an external perturbation, such as heating and cooling, is applied, resulting in crystallization. The phase transition from crystalline solid to metastable liquid can be attributed to the lower interfacial energy in liquid/solid interface than that in crystal/crystal interface. Our results provide direct evidence of the existence of the metastable liquid as an intermediate state in solid–solid phase transitions.

## Methods

**Sample configuration.** Symmetric DACs with 300–500 μm anvil culets were used for high-pressure and high-temperature experiments. Under hydrostatic condition with neon as pressure medium, a small piece of Bi sample (Alfa Aesar, purity of 99.99%) with typical dimensions of 30–40 μm in diameter and ∼20 μm thick was loaded into the centre of the sample chamber on the piston side of the DAC. The diameter of the sample chamber was ∼150 μm in a rhenium gasket with a typical indentation thickness of ∼60 μm. A small pressure marker, consisting of a mixture of NaCl and MgO, was loaded on the cylinder side of the DAC[35], together with a ruby sphere used in the gas loading process to control the starting pressure[36]. The mixture of NaCl and MgO was used to prevent NaCl from growing into a single crystal. The pressure was determined using the equation of state of NaCl[35], which gives a high precision in pressure measurements of <0.1 GPa. Neon was loaded as pressure medium using the gas-loading system of GeoSoiEnviroCARS[37]. The Bi sample and the mixture of NaCl and MgO were insulated by pressure medium of Ne. Several experiments were conducted under nonhydrostatic conditions, where the sample chamber was either filled with the Bi sample together with a small piece of NaCl marker[35], or a chip of the Bi sample surrounded by NaCl, which served as both pressure medium and marker.

**Temperature measurement and pressure control.** Loaded DACs were placed in a whole-cell heater assembly for high-temperature conditions[38]. Two thermocouples were used to measure the temperatures with one attached to the holder near the heater and the other attached to the gasket in the cell. The temperature difference was small, less than 2 K. Therefore, the thermal gradient in the DAC was minimal because the cell is heated uniformly. Double gas membranes[38] were used to control compression and decompression pathways precisely. To avoid the leakage of neon at high temperature, the sample was

compressed to ∼1 GPa before heating. During the compression and decompression process, the temperature was very stable within ±1 K, as shown in Supplementary Fig. 3.

**High-pressure X-ray diffraction measurement.** *In situ* angle dispersive X-ray diffraction was performed at beamlines 16-ID-B and 16-BM-D at HPCAT at the Advanced Photon Source, Argonne National Laboratory. The X-ray beam with wavelength of 0.4066, 0.3066 or 0.2480 Å was used and focused into a $5 \times 6 \, \mu m^2$ (full width at half maximum, FWHM) spot on the sample[39,40]. Two-dimensional (2D) diffraction images were collected with a PILATUS 1M-F detector or Mar345 Image Plate. The typical exposure time was 10 s at beamline 16-ID-B, and 60 s at 16-BM-D. 2D diffraction images were integrated using the Dioptas or Fit2D software[41,42]. The structure of liquid Bi was analysed using the pdfgetx software[43], from which the structure factors and the pair distribution functions at different pressures were obtained.

**Optical imaging measurements.** We also used optical imaging to monitor the sample morphology under compression and decompression processes under hydrostatic condition at 489 K by using a high-speed camera (Photron FASTCAM SA3) at HPCAT. The frame rate in Supplementary Movies is 125 f.p.s. (frames per second). The results can be viewed in movies in the Supplementary Materials.

**Data availability.** The data that support the findings of this study are available from the corresponding author upon request.

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

## Acknowledgements

We thank Yanbin Wang for helpful discussion. This research was supported by DOE-BES, Division of Materials Sciences and Engineering under Award DE-FG02-99ER45775, and DOE-NNSA, Stewardship Science Academic Programs under Award DE-NA0001974. HPCAT operations are supported by DOE-NNSA under Award No. DE-NA0001974 and DOE-BES under Award No. DE-FG02-99ER45775, with partial instrumentation funding by NSF. The Advanced Photon Source is a U.S. Department of Energy (DOE) Office of Science User Facility operated for the DOE Office of Science by Argonne National Laboratory under Contract No. DE-AC02-06CH11357. We thank the assistance of Sergy Tchakev in gas-loading samples. Use of the COMPRES-GSECARS gas-loading system was supported by COMPRES under NSF Cooperative Agreement EAR 11–57758 and by GSECARS through NSF grant EAR-1128799 and DOE grant DE-FG02-94ER14466.

## Author contributions

G.S. and C.L. designed the experiments; C.L., J.S.S., S.V.S., Y.K., C.P. and C.K.-B. performed the experiments; C.L. analysed the data; C.L. and G.S. interpreted the data and wrote the paper with contributions from all the co-authors.

## Additional information

**Competing financial interests:** The authors declare no competing financial interests.

**Publisher's note**: 

