## [Peer Review File · Nature Communications]

Reviewers' comments:

Reviewer #1 (Remarks to the Author):

The manuscript reports on the novel phenomenon of the formation of a metastable liquid from the solid state. It is the first time that such effect is shown for a variation of pressure. The authors should highlight early in their work this novel relation to pressure (maybe also in the title) since the melting to a metastable liquid has been recently demonstrated experimentally for heating of colloidal model system and also in metals (this papers are correctly cited). However, in this two recent studies the metastable liquid could not be retained for a significant time, which was archived in the submitted work and enable for a detailed study (e.g. as demonstrated via X-ray experiments). Moreover, the work is important, since the experimental verification of the phenomenon needs definitely more support and the pressure effect in the paper indicates a universality of the concept of metastable melting. The paper is sound, represents very interesting work important to a broad audience and should definitely be published.

Beside the above-mentioned issues, there are only minor points, which should be addressed to improve the paper. The position of the suggested amendments is indicated in the annotated pdf.

- [1] This has been already shown for metallic glasses upon heating. Please correct for this.
- [2] Structure should be changed according to the guidelines of NC. Please include more from the SI in the main text. That would make the paper much easier to read.
- [3] Please give an explanation for this somewhat "unexpected" effect for the reader who is not an expert for Bi (volume change?). Why is there a difference to hydrostatic conditions?
- [4] Why is there such a difference? This needs to be explained.
- [5] Was this done for different samples? Is there any effect of the used medium? Please comment this.
- [6] DIL is also used for dilatometry, and might be changed.
- [7] Please explain this briefly.
- [8] Why? Please add a comment to guide the reader.
- [9] It would be nice to have a plain equilibrium phase diagram of Bi from literature in the SI to help the reader.
- [10] The authors should address in a clearer way of what nature the thermodynamic driving force is during the transition. This needs already here a bit more attention to guide the reader.
- [11] This is not very intuitive. Maybe not all data need to be presented to make it easier to read the figure.
- [12] Please indicate Bi-III'.

Table S1: Please add references.

Reviewer #2 (Remarks to the Author):

This review is in response to the article "A metastable liquid melted from a crystalline solid" by Lin et al. Overall this submission addresses the novel formation of a proposed metastable liquid with decompression and other external stimuli. To my knowledge, there are not many decompression related publications, as it typically serves as a spot-check for compression results. As such the results of this work are more broadly interesting for a number of fields. In particular, this would likely be a useful consideration for planetary formation, geoscience, and materials science, all of which are likely to deal with systems in such regimes of pressure and temperature. While the peculiarities of bismuth are interesting, gallium also has a similar lowering of melt temperature with pressure in its phase diagram as well, and these results may be used in such studies as well. As such, it is undoubtedly a novel result and would be of broad interest. However, I have a few concerns that I feel would help to bolster the argument.

- The authors state that the temperature is stable to "within +/- 1K." Do you have a temperature trace that can be added to the supplementary material in support of this? It would be very helpful to assuage readers that random thermal fluctuations are not responsible for some of your results. On a related note, do you have information as to the pressure gradients present? As both NaCl and MgO were used immediately surrounding the sample, it would seem that this should be measureable and would help to show a consistent stress state for the sample (despite the hydrostaticity of neon).

- There may be a terminology issue involved with the use of "cracking" in conjunction with video 1. While I definitely agree that there is a dramatic change in the sample, I don't know that cracking is the appropriate description of this. It appears to be melting to me. As such, is there other evidence in support of the cracking proposed?

- On a more technical note, the authors state that they cycled the compression/decompression pathway five times for the 489K study. Was this done for the other temperatures as well? Additionally, was the same sample used or was it a pristine sample each time?

- Have the authors used the PDF traces in their supplementary information to obtain any information about the meta-stable liquid? It would seem that a model for the liquid could be obtained from this to help theoretical modellers to "reproduce" such results and would be of great benefit to the community in understanding this phenomenon.

Beyond just these suggestions, there are several places where English usage becomes an issue. I would suggest a careful proof-read to ensure such issues are caught. With these issues address, I would recommend publication.

Reviewer #3 (Remarks to the Author):

The manuscript presents results from an experimental study investigating the metastable phase behavior of elemental bismuth below its melting line. Using in situ x-ray diffraction, the authors probe the nature of the pressure-induced solid-solid transition from Bi-I to Bi-IV. Isothermal compression of Bi-I to Bi-IV occurs via a multistep mechanisms involving sequential transitions from Bi-I to Bi-II and Bi-II to Bi-IV. During decompression of Bi-IV, however, a metastable liquid intermediate forms prior to the crystallization of Bi-I. The authors show evidence that this metastable liquid can be maintained for hours without interference from crystallization.

The experimental findings reported in the manuscript demonstrate the formation of a metastable liquid from a crystalline solid at conditions below the normal melting temperature. Hints of similar unusual behavior have been reported in colloidal systems and metallic melts. In the referee's opinion, however, this is the first convincing experimental evidence. Furthermore, the fact that the metastable liquid can be observed for long durations opens new possibilities for understanding the role of metastable intermediates in solid-solid transitions. The study is technically sound and provides significant insights that will be of interest to the readership of Nat. Comm. I am therefore able to recommend the manuscript for publication.

Response to review's comments

Reviewer #1 (Remarks to the Author):

The manuscript reports on the novel phenomenon of the formation of a metastable liquid from the solid state. It is the first time that such effect is shown for a variation of pressure. The authors should highlight early in their work this novel relation to pressure (maybe also in the title) since the melting to a metastable liquid has been recently demonstrated experimentally for heating of colloidal model system and also in metals (this papers are correctly cited). However, in this two recent studies the metastable liquid could not be retained for a significant time, which was archived in the submitted work and enable for a detailed study (e.g. as demonstrated via X-ray experiments). Moreover, the work is important, since the experimental verification of the phenomenon needs definitely more support and the pressure effect in the paper indicates a universality of the concept of metastable melting. The paper is sound, represents very interesting work important to a broad audience and should definitely be published.

Response to the summary comments of Review #1:

We appreciate the positive comments on the novelty and importance of this work. We agree that the pressure aspect needs to be highlighted, and have modified the title to be “*A metastable liquid melted from a crystalline solid under decompression*”.

Beside the above-mentioned issues, there are only minor points, which should be addressed to improve the paper. The position of the suggested amendments is indicated in the annotated pdf.

[1] *This has been already shown for metallic glasses upon heating. Please correct for this.*

Response:

Thanks for mentioning this. We have replaced the sentence by the following:
“*Recently, there has been a growing interest⁴⁻¹¹ in studying whether a crystalline solid may directly melt into a metastable liquid below the melting line (T_m) and the mechanisms underlying it*”.

[2] *Structure should be changed according to the guidelines of NC. Please include more from the SI in the main text. That would make the paper much easier to read.*

Response:

The structure of the manuscript has been revised according to the guideline of NC.

As suggested, we have added a paragraph (as shown below) to describe the phase diagram of Bi and crystal structure from Fig. S1 in the main text, which will make the paper much easier to read:

“**Phase diagram.** *Bismuth has a complex phase diagram, exhibiting several polymorphs and a V-shape melting curve (Supplementary Fig. 1)¹⁴. At ambient conditions, the rhombohedral structure (Bi-I) is the stable phase with $Z=2$ and space group $R-3m$ (Supplementary Fig. 1). Bi-I*

melts at ~ 544 K and ambient pressure¹⁴. The structure of Bi-I can be viewed as a slightly distorted primitive cubic structure¹⁵. Similar to ice Ih, Bi-I has a negative Clausius-Clapeyron melting slope. Under compression at room temperature, Bi-I transforms to Bi-II with volume collapse of $\sim 4.7\%$ at ~ 2.5 GPa¹⁴. Bi-II has a monoclinic structure with space group P21/n and Z=8 (Supplementary Fig. 1)¹⁶. The layer structure of Bi-II is similar to Bi-I, and can be described as a heavily distorted primitive cubic array¹⁵. Upon further compression, Bi-II transforms to Bi-III at ~ 2.8 GPa¹⁷, a tetrahedral host-guest structure (Supplementary Fig. 1). Bi-II' was found at 1.9 GPa and 463 K and exists in a small pressure-temperature region¹⁸. It has the β -tin type structure with $a=6.210$ Å and $b=3.311$ Å. The triple points are at 1.65 GPa and 465 K for I-II'-liquid, 1.96 GPa and 465 K for II-II'-liquid, and 2.1 GPa and 464 K for II-IV-liquid^{14,18}. Bi-II' and Bi-II have a flat melting line. Above 465 K, liquid Bi transformed to Bi-IV under compression¹⁴. The high-pressure and high-temperature phase of Bi-IV has an orthorhombic oC16 structure with $a=11.191(5)$ Å, $b=6.622(1)$ Å and $c=6.608(1)$ Å¹⁹, as shown in Supplementary Fig. 1.”

[3] Please give an explanation for this somewhat “unexpected” effect for the reader who is not an expert for Bi (volume change?). Why is there a difference to hydrostatic conditions?

Response:

Bi-I has a negative melting line. Analogous to ice, the volume change between liquid and solid is negative upon its melting. This leads to negative Clausius-Clapeyron melting slope and decrease of melting temperature with increasing pressures. We have explained this in the phase diagram of Bi in the revised manuscript, also in the response of comments [2].

Shear stress could play an important role in melting of Bi-I, as found in some other materials as well (see Ref. 20-24 in the revised manuscript). Under hydrostatic condition using neon as pressure transmitting medium, we observed clear crystalline-crystalline transitions at temperatures below the melting temperature. However, under nonhydrostatic condition (no pressure medium or solid NaCl as medium), we observed the melting of Bi-I at similar temperatures, indicating a reduction of melting temperature of Bi-I due to stress effect. We have briefly explained this in the revision.

[4] Why is there such a difference? This needs to be explained.

Response:

Yes, we have explained this briefly in the revised manuscript: “Apparently, the shear stress may have played an important role in the phase transition of Bi-I, as reported in other materials²⁰⁻²⁴. The use of liquid neon in this study provides excellent hydrostatic condition, thus

revealing the true melting temperature higher than 489 K.” Please also see the response to the comment [3].

[5] Was this done for different samples? Is there any effect of the used medium? Please comment this.

Response:

Yes, a fresh sample was used in each experimental run. We have clarified this in the revision (Page 3).

Here, neon was used as pressure medium to provide a hydrostatic condition and to avoid stress effect on the phase transition of Bi. There is no other particular effect of neon on phase transition, *e.g.* neon diffusion into Bi. This is because the atomic radius of neon is much larger than the interstitial of the crystalline structure of Bi-I. Moreover, we have checked the structure of sample at room temperature after high P-T experiments, and did not find any difference in the crystal structure or unit cell parameters compared with those of starting materials.

[6] DIL is also used for dilatometry, and might be changed.

Response:

In this manuscript, nothing is involved with dilatometry. We would like to keep the abbreviated “DIL” for decompression-induced liquid; it is convenient for this manuscript and we believe that it will not make much confusions to readers.

[7] Please explain this briefly.

Response:

This is an experimental observation. The melting pressures represent the pressure values where the metastable liquid starts to crystalize under decompression pathways at various temperatures. The melting pressure is found to decrease with reducing temperatures.

[8] Why? Please add a comment to guide the reader.

Response:

There is only one thermodynamic equilibrium state at a given condition (*i.e.*, pressure and temperature). If two phases are observed at a certain pressure and temperature condition, it means that at least one of them is not a thermodynamic equilibrium state, *i.e.*, one of them must be a metastable phase.

[9] It would be nice to have a plain equilibrium phase diagram of Bi from literature in the SI to help the reader.

Response:

We have added description of the phase diagram of Bi and clear references in the main text to help the reader (Please see comment [2]). Because a phase diagram is already included in the main text (See Fig. 2), we would like to avoid the repetition of a similar diagram in the SI.

[10] *The authors should address in a clearer way of what nature the thermodynamic driving force is during the transition. This needs already here a bit more attention to guide the reader.*

Response:

Thanks for pointing out that. In page 6 of the revised manuscript, we clearly state that the thermodynamic driving force during the phase transition comes from the difference of Gibbs free energy between parent and new phase.

[11] *This is not very intuitive. Maybe not all data need to be presented to make it easier to read the figure.*

Response:

In fact, all the results and discussion are based on these experimental data presented in Fig. 2. These results lead to conclude the observation of metastable liquid by decompression, metastability of DIL by cooling and heating, and supercooled liquid. We think the present figure provides a good summary to display our main results.

[12] *Please indicate Bi-III'.*

Response:

In revised Fig. 2, we have indicated Bi-III' (red open down triangle: ∇) during the decompression process.

Table S1: Please add references.

Response:

We calculated the lattice parameters using our x-ray diffraction data. We have added the references from which the structure model we used.

Reviewer #2 (Remarks to the Author):

This review is in response to the article "A metastable liquid melted from a crystalline solid" by Lin et al. Overall this submission addresses the novel formation of a proposed metastable liquid with decompression and other external stimuli. To my knowledge, there are not many decompression related publications, as it typically serves as a spot-check for compression results. As such the results of this work are more broadly interesting for a number of fields. In particular, this would likely be a useful consideration for planetary formation, geoscience, and materials science, all of which are likely to deal with systems in such regimes of pressure and temperature. While the peculiarities of bismuth are interesting, gallium also has a similar lowering of melt temperature with pressure in its

phase diagram as well, and these results may be used in such studies as well. As such, it is undoubtedly a novel result and would be of broad interest. However, I have a few concerns that I feel would help to bolster the argument.

Response to the summary comment of Review #2:

We appreciate the positive comments.

Comment 1:- The authors state that the temperature is stable to "within +/- 1K." Do you have a temperature trace that can be added to the supplementary material in support of this? It would be very helpful to assuage readers that random thermal fluctuations are not responsible for some of your results.

Response to comments:

Yes, we have traced temperatures on every runs. A typical plot has now been added as a supplementary figure Fig. S3B (also shown below).

Comments 2: On a related note, do you have information as to the pressure gradients present? As both NaCl and MgO were used immediately surrounding the sample, it would seem that this should be measureable and would help to show a consistent stress state for the sample (despite the hydrostaticity of neon).

Response:

The Bi sample in our study was surrounded by neon, and well separated from NaCl and MgO (See an image below). During the sample preparation, the Bi sample was loaded on the piston side of DAC while a small piece of mixed NaCl and MgO was loaded on cylinder side, as shown in the supplementary Fig. 3A (also shown below). From optical inspection, we can make sure that MgO and NaCl were well separated from the Bi sample.

Supplementary Fig. S3

Comment 3: - *There may be a terminology issue involved with the use of "cracking" in conjunction with video 1. While I definitely agree that there is a dramatic change in the sample, I don't know that cracking is the appropriate description of this. It appears to be melting to me. As such, is there other evidence in support of the cracking proposed?*

Response:

We agree with the reviewer that the use of “cracking” may not be an appropriate description of Video 1. In Video 1, there are clear morphology changes with compression; in the end there remained sharp edges of irregular shapes after the change. In Video 2, the clear morphology change is dominated by the edge smoothing, leading to more round edge textures. We interpret the edge smoothing arising from the surface tension of liquid, while the irregular sharp edges representing solid-solid phase transition(s). The later interpretation is clearly supported by our x-ray diffraction data. We have modified the descriptions of Video 1 and Video 2 in the supplement.

Comment 4: *On a more technical note, the authors state that they cycled the compression/decompression pathway five times for the 489K study. Was this done for the other temperatures as well? Additionally, was the same sample used or was it a pristine sample each time?*

Response:

We cycled the compression/decompression pathway five times only at 489 K. Initially, we wanted to check if there is any effects from different compression and decompression rates. On the fourth and fifth times, we just wanted to confirm the surprising results of metastability. Fresh samples were used every time at 489 K. For runs at other temperatures, we repeated compression/decompression pathway only once or twice. In each run, a fresh sample was always used. To be more specific, we carried out experiments once for 482 K and 441 K, twice for 455 K. We have added the information in table S2.

Comment 5: *Have the authors used the PDF traces in their supplementary information to obtain any information about the meta-stable liquid? It would seem that a model for the liquid could be obtained from this to help theoretical modellers to "reproduce" such results and would be of great benefit to the community in understanding this phenomenon.*

Response:

Yes. The results are included in Figure S7A and B for the evolution of the structure factor ($S(Q)$) and the reduced pair distribution function ($G(r)$) of metastable liquid with pressures. Figure 4a shows the changes of nearest-neighbor coordination number with pressures.

Beyond just these suggestions, there are several places where English usage becomes an issue. I would suggest a careful proof-read to ensure such issues are caught. With these issues address, I would recommend publication.

Response:

We have made a careful proof-read of the revised manuscript by a native English speaker.

Reviewer #3 (Remarks to the Author):

The manuscript presents results from an experimental study investigating the metastable phase behavior of elemental bismuth below its melting line. Using in situ x-ray diffraction, the authors probe the nature of the pressure-induced solid-solid transition from Bi-I to Bi-IV. Isothermal compression of Bi-I to Bi-IV occurs via a multistep mechanisms involving sequential transitions from Bi-I to Bi-II and Bi-II to Bi-IV. During decompression of Bi-IV, however, a metastable liquid intermediate forms prior to the crystallization of Bi-I. The authors show evidence that this metastable liquid can be maintained for hours without interference from crystallization.

The experimental findings reported in the manuscript demonstrate the formation of a metastable liquid from a crystalline solid at conditions below the normal melting temperature. Hints of similar unusual behavior have been reported in colloidal systems and metallic melts. In the referee's opinion, however, this is the first convincing experimental evidence. Furthermore, the fact that the metastable liquid can be observed for long durations opens new possibilities for understanding the role of metastable intermediates in solid-solid transitions. The study is technically sound and provides significant insights that will be of interest to the readership of Nat. Comm. I am therefore able to recommend the manuscript for publication.

Response to the summary comments of Review #3:

We appreciate the reviewer's positive comments.

REVIEWERS' COMMENTS:

Reviewer #1 (Remarks to the Author):

The authors have considered the most important suggestions made. They could have included more data from the SI into the main text, but the paper can also be published in its present form in NC. Overall, it's nice work!

Reviewer #2 (Remarks to the Author):

I feel that all of my questions regarding the paper and as such I would recommend publication. Very nice work!

REVIEWERS' COMMENTS:

Reviewer #1 (Remarks to the Author):

The authors have considered the most important suggestions made. They could have included more data from the SI into the main text, but the paper can also be published in its present form in NC. Overall, it's nice work!

Response. Thanks for the positive comments. We prefer to keep the present version.

Reviewer #2 (Remarks to the Author):

I feel that all of my questions regarding the paper and as such I would recommend publication. Very nice work!

Response. Thanks for the positive comments.